# Fabrication of Ultra-Sharp Tips by Dynamic Chemical Etching Process for Scanning Near-Field Microwave Microscopy

**DOI:** 10.3390/s23063360

**Published:** 2023-03-22

**Authors:** C. H. Joseph, Giovanni Capoccia, Andrea Lucibello, Emanuela Proietti, Giovanni Maria Sardi, Giancarlo Bartolucci, Romolo Marcelli

**Affiliations:** 1Institute for Microelectronics and Microsystems, National Research Council (CNR-IMM), Via del Fosso del Cavaliere 100, 00133 Rome, Italy; 2Department of Electronic Engineering, University of Rome "Tor Vergata", Via del Politecnico 1, 00133 Rome, Italy

**Keywords:** scanning near-field microwave microscopy (SNMM), tapered probes, dynamic chemical etching, microwave imaging

## Abstract

This work details an effective dynamic chemical etching technique to fabricate ultra-sharp tips for Scanning Near-Field Microwave Microscopy (SNMM). The protruded cylindrical part of the inner conductor in a commercial SMA (Sub Miniature A) coaxial connector is tapered by a dynamic chemical etching process using ferric chloride. The technique is optimized to fabricate ultra-sharp probe tips with controllable shapes and tapered down to have a radius of tip apex around ∼1 μm. The detailed optimization facilitated the fabrication of reproducible high-quality probes suitable for non-contact SNMM operation. A simple analytical model is also presented to better describe the dynamics of the tip formation. The near-field characteristics of the tips are evaluated by finite element method (FEM) based electromagnetic simulations and the performance of the probes has been validated experimentally by means of imaging a metal-dielectric sample using the in-house scanning near-field microwave microscopy system.

## 1. Introduction

Probes are the most important features in any scanning probe microscopy system and particularly in near-field microwave microscopy where the geometry of the probe tip plays an important role in defining the spatial resolution as well as the sensitivity of the microwave microscopy setup [1,2]. Taking advantage of the near-field or evanescent field in microscopy systems has enabled researchers to overcome the classical diffraction limit in the spatial resolution of the microscopy [3,4]. This near-field condition can be achieved by bringing the probe/source closer to the sample surface where the distance between them becomes much smaller than that of the free space wavelength of the irradiated electromagnetic wave [5]. This near-field technique has been used in optical and microwave regimes to achieve ultramicroscopic resolution [3,4]. Scanning near-field microwave microscopy (SNMM) is one of the types of scanning probe microscopy that facilitates nanometric characterization of materials in terms of electrical properties at a microwave frequency range where these materials are used to be operated widely in nanoscale devices. This makes SNMM appealing to a broad range of applications in measuring electrical quantities for different materials such as dielectric [6], semiconductors [7,8], 2D materials [9,10], ferroelectric [11,12], ferromagnetic [13], polymer composite materials [14] and even biological systems [15,16,17]. Another advantage of using microwaves lies in their ability to penetrate through dielectric and low conductive materials and see the subsurface characteristics in a non-destructive way [18,19].

Typically, near-field probes are classified into two types: aperture-based probes and aperture-less probes. The concept of near-field microscopy was developed first based on the aperture-based probes by Synge [3] and followed by several others [4,20,21]. The aperture-based probes have a significant drawback of a low radiation throughput as the incident wave have to pass through a cut-off region which limits the sensitivity of these type of probes and the resolution is highly influenced by the aperture size [22]. On the other hand, the aperture-less probes are mainly field-concentrating designs using sharp tapered tips such as scanning tunneling microscopy (STM), atomic force microscopy (AFM) or other coaxial-based designs to highly localize the electric fields at the tip apex and strongly confine the field interaction with the sample [5]. The resolution of the probe can be improved by sharpening the end of the tapered tip. A sharp tip will act as a tiny antenna emitting the microwave signal and then collecting the reflected signal from the sample [23,24].

Near-field microwave probes can be made to work either in broadband or resonant modality. In the resonant case, the measurement’s sensitivity is enhanced but works in a narrow band. The SNMM probe designs have advanced over the years with the development of different probe designs, including strip lines [25], microstrip lines [26], and open-ended coaxial resonant probes [27]. Coaxial-based probes have several advantages over other waveguiding structures because there are no cut-off wavelength limits and the outer conductor shields and properly guides the signal [22]. Many different coaxial-based feeding techniques are applied to silicon-made AFM cantilevers, which also involve complicated microfabrication techniques. However, these probes have a crucial drawback, i.e., the chip body and the cantilever parts contribute to strong parasitic capacitances induced by the capacitive coupling to the measurements, which significantly affect the desired tip-sample interaction (in the order of ∼1 aF). To prevent the strong cross-talk coming from different parts of the probe, many researchers worked on several designs, such as shielding the AFM cantilever [28], increasing the height of the probe tip [29], and adding a nanowire tip on the top of the silicon probe [30].

Flanged or non-flanged open coaxial probes have long been used in conventional microwave material characterization techniques [31]. SMA connector-based coaxial probes used in near-field microwave microscopy systems have been reported in homemade setups [32] and also in transmission-based microwave microscopy systems to irradiate the sample from the bottom, thus increasing the sensitivity of the measurement compared to the reflection measurement [33].

This work presents a step-by-step process of tapering the protruded part of the inner conductor pin of a broadband microwave flange connector to be utilized in a high-resolution SNMM system. Finally, a comparison between resolutions obtained with and without etching the tip in imaging metal-patterned shapes manufactured onto an alumina substrate is presented and discussed. This paper proposes a cost-effective solution for tips to be used in high-frequency near-field imaging when both a wide scanning area (up to several mm${}^{2}$) and high resolution are needed.

## 2. Materials and Methods

### 2.1. Chemical Etching Setup

The standard commercial SMA flange connectors used in this work are made up of non-magnetic stainless steel passivated bodies with gold plated finish [34]. The inner conductor of the SMA is beryllium copper and the dielectric part is PTFE (polytetrafluoro ethylene) also called Teflon as a commercial product. The SMA connector with protruded PTFE configuration is utilized with the intention to avoid the metal flange having any direct interaction with the etching chemical solution. The total protruded length of the SMA connector pin is 3.2 mm with PTFE covering the pin outside the flange up to 1.6 mm. The extended cylindrical pin has a diameter of 250 μm. The geometry of the female jack SMA connector is shown in Figure 1a. In the dynamic etching process, the protruding part of the pin goes up and down inside the etching solution. During the process, the end part of the pin stays longer in the solution than the top end thus creating a tapered conical shape tip. The developed experimental arrangement for the controlled movement of the pin inside the etching solution comprises a stepper motor and a customized stage arrangement to fix the pin stability. The stage and experimental arrangement are shown in Figure 1b,c. The moving stage has been made up of a plastic material and fabricated using 3D printing technology. An SMA male plug is attached at the end of the customized stage to connect the female jack SMA to be etched.

The 3D-printed moving stage is attached to the stepper motor. The stepper motor has a precision of 5 μm per step movement. Two digital USB microscopic cameras are mounted in two viewing positions to monitor the complete etching process. Custom-made software has been developed to control the movement with all the specific input parameters required for the dynamic etching process. The first step of the process is to fix the zero line where the pin touches the etching solution with the help of the digital microscopy camera. After fixing the zero line, the insertion depth as well as the release height should be given as input into the software. Based on the input parameters which include the number of dipping steps and the velocity, a G-code will be generated automatically from the software. The generated G-code will then be imported into the software to control the stepper motor movement. In this way, the whole process is made fully automated.

### 2.2. Chemical Etching Process

The central conductor pin of the commercial SMA connectors has a gold-plated finish with a thickness of ∼0.25 μm. The dynamic chemical etching process mainly involves two steps: (1) removal of the plated gold on the center conductor pin with a gold etchant solution and (2) dynamic etching of the beryllium copper pin with ferric chloride (FeCl_3_) solution.

The most common etchant solution useful for gold etching is the iodine-iodide system. This solution is typically formulated from potassium iodide (KI) or sodium iodide (NaI) salts and iodine (I_2_). As a first step of the etching procedure, a potassium tri-iodide (KI_3_) solution has been prepared in two different levels of concentration to remove the gold coating from the pin. The first solution has a concentration of 5 g of KI and 1.21 g of iodide prepared in 200 mL of water. The second solution was prepared with a little higher concentration of 20 g of KI and 5 g of iodide in 200 mL of water. The protruding pin has been immersed in the prepared solution and left therein for some time until the gold coating is completely etched away from the surface of the pin. The higher concentrated solution takes approximately 10 and 15 min to completely etch the gold coating, while the lower concentrated solution takes around 30 min. After etching the gold coating, a microscopic visual inspection is necessary to identify if there are any gold residuals settled on the surface of the pin. This may affect the smooth etching of copper during the dynamic etching process and thus resulting in asymmetries in the final shape of the tapered tips. Furthermore, a higher number of cycles in the dynamic copper etching process may require to reduce the asymmetry in the shape caused by these gold residuals. The next step in the process is to etch the inner beryllium copper to shape the cylindrical pin into the tapered conical tip. Plenty of methods for etching copper were reported in the kinds of literature over the past years. The vital part of the developed dynamic etching is to reduce the bubbles around the pin during the process and have good control over the etching process. For those reasons, we have adopted ferric chloride (FeCl_3_) solution as an etchant for copper and avoided using more aggressive etchants such as aqua regia. As per the basic chemical reactions of ferric chloride with copper, it dissolves effectively without producing any gas which in turn creates no bubbles due to the chemical effects. The commercial FeCl_3_ etchant solution generally comes with a concentration between 30% to 40%. To optimize the suitable concentration for the dynamic etching process, the solution with four different concentrations as 20% (10 g/25 mL), 25% (7.5 g/25 mL), 30% (6.25 g/25 mL), and 40% (5 g/25 mL), were prepared. Moreover, the room temperature was imposed to avoid an uncontrolled process acceleration and to improve the result with a smooth lateral shape of the etched tip.

### 2.3. Tip Formation Dynamics

The tapering of the metal pin has to be performed by using a properly calibrated etching solution as mentioned above, which will affect the final shape of the pin depending on many parameters. Starting from the geometry shown in Figure 2, the full connector will be moved up and down with a speed of V(t), and the pin will be tapered accounting for the specific time dependence imposed on the dipping velocity.

In the geometry, the flange is followed by a dielectric material with a length *D*, and the pin with full length *L*. For improving the etching control, a small distance ε should separate the dielectric from the pin. δ is a quantity to be defined, related to meniscus effects close to the surface of the liquid (etching) solution, which may influence the shape at the end of the pin. The full process flow of insertion and release is illustrated in Figure 3. The resulting shape of the pin after the etching is qualitatively shown in Figure 4a,b. Meniscus effects manifest themselves when the end of the pin is close to the liquid surface; in this case, the dipping begins before the full immersion as well as when the pin is going out from the etching solution. Increasing the release height is essential to avoid any solution sticking with the pin.

The pin, entering the liquid solution, experiences an etching as a function of the time *t* given by:(1)$$S{}_{e}\left(t\right)=r{}_{e}\left(T\right)\times t$$
where $S{}_{e}\left(t\right)$ is the etched portion of the pin on one side and, $r{}_{e}\left(T\right)$ is the etching rate at a given temperature *T*. The room temperature was chosen to control the etching process better, as discussed in [35]. The situation is schematized in Figure 5.

Because of the symmetry of the problem and the isotropy of the etching, it is possible to treat the etching model in 2D. In particular, the width of the pin will evolve as a function of time in such a way that:(2)$$w\left(t\right)=W-2\times S{}_{e}\left(t\right)=W-2\times r{}_{e}\left(T\right)\times t$$

The end of the pin will be fully etched when:(3)$$W=2\times r{}_{e}\left(T\right)\times N\times 2t{}_{0}$$
where $t_{0}$ is the single dipping time, to be doubled for considering the double travel inside the etching solution, *T* is the temperature and *N* is the number of cycles to be imposed. In fact, to obtain a sharp conical-shaped tip as a final form, the pin has to be dipped and taken out for a number of cycles that depends on the concentration of the solution, with the result of a more pronounced etching at the end of the pin. To get w=0, we can say that:(4)$$t_{etching}=\frac{W}{2\times r{}_{e}\left(T\right)}=2\times N\times t_{0}$$
(5)$$N=\frac{W}{4\times r_{e}\left(T\right)\times t_{0}}$$

Two time variables should describe the etching process, $t_{e}$ for etching along the x-direction, normal to the pin, and $t_{0}$ for dipping along the z-direction, parallel to the pin. On the other hand, no etching will happen if dipping is not present, unless we correct for the meniscus effect, and we can use just the *t* variable to combine both processes and predict the etched final shape of the pin.

Then, we can write:(6)$$w\left(t\right)=W-2\times r_{e}\left(T\right)\times t$$
(7)$$z\left(t\right)=v_{0}\times t$$
or
(8)$$z\left(t\right)=v_{0}+\frac{1}{2}at^{2}$$

The last two motion equations take into account a constant speed in dipping or for an accelerated regime. Because of the two possible solutions, and assuming for the moment no effects due to meniscus, the shape of the etched pin as a function of *z* can be obtained by solving for t=t(z), and:(9)$$w\left(z\right){|}_{Const.speed}\;=W-2r_{e}\left(T\right)\times t\times \frac{z}{v_{0}}$$
(10)$$w\left(z\right){|}_{Accel.speed}\;=W-2r_{e}\left(T\right)\times \frac{\sqrt{v_{0}^{2}+2az}-v_{0}}{a}$$

An above analytical solution is certainly a zero-order approach and not accounting for all the possible experimental boundary conditions, but it is significant because of the theoretical possibility to shape the pin depending on the imposed speed regime during the etching process. In this case, additional attempts can be performed to achieve an arbitrary shape, possibly including integrated resonating geometries useful for effective radiation and electrical matching of the high-frequency signal.

## 3. Results and Discussion

### 3.1. Dynamic Etching

To explore more in detail the optimal parameters for the dynamic etching process, the velocity of the pin movement inside the etchant solution and the concentration of the etchant solution has been varied. Prior to starting the copper etching procedure, all the SMA flange protruded pins were processed with potassium triiodide (KI_3_) solution to remove the gold coating.

To optimize the velocity parameter, three different velocities have been considered 1 mm/s, 2 mm/s and 4 mm/s, respectively. The number of movement cycles as well as the concentration of the etching solution has been kept unchanged for all the velocity variations. The concentration of the solution was maintained at 25%. The whole process was carried out in several steps with 100 cycles each to monitor the reshaping of the tip. The time taken for every 100 cycles has been noted and added to the end to get the total etching time. The velocity optimization parameters used for the etching process are listed in Table 1.

The complete etching process was monitored by the digital microscopic cameras and the etching process in different intervals of cycles is shown in Figure 6. It is observed from this optimization that the tips produced with a velocity of 2 mm/s yield better quality compared to the other two velocities. The process with a velocity of 2 mm/s can be able to provide perfectly tapered tips with a cone angle of less than 15° and the sharpness of the tip apex reaches to sub-micron scale with a well-controlled etching.

As shown in Figure 6a–f, the tapering of the pin started to be visible after 300 cycles and increases further with a higher number of cycles. The sharp tapered tip with the desired quality was achieved around 1000 cycles.

The next set of optimizations was performed by varying the concentration of the etchant ferric chloride solution. Figure 7 shows the photograph of the prepared etchant solutions with a concentration of 20%, 25%, 30%, and 40%, respectively. The parameters related to the optimization process are listed in Table 2. Both the velocity and insertion depth has been kept constant for this set of optimizations to have a reasonable comparison between the different concentrations.

The final shapes of the tips after the etching process with different concentrations are shown in Figure 8. The highly concentrated solutions with 30% and 40% take less number of cycles to have sharper tips but also have less control over the etching. It is evident from Figure 8a–d, that the lower concentrated solutions with 20% and 25% produce uniformly tapered high-quality tips while the highly concentrated solutions produce not perfectly tapered tips with high roughness.

Figure 8c also reveals a gold residual sitting on the surface of the pin. As the chosen insertion depth is below the residual, any asymmetry in the final shape of the tip tapering was avoided. The detailed inspection of the optimization process indicates the best-optimized parameters for obtaining the better-quality tips in a more controlled way is to keep the velocity as well as the concentration to be fixed at 2 mm/s and 25%, respectively.

The tip morphology was analyzed with the help of Scanning Electron Microscopy (SEM) and it reveals that the final tip apex is having smaller diameters of around ∼1 μm and showed a nice conical shape. The angle of the tip was further reduced by increasing the insertion depth and obtaining a cone angle below 15°. The cone angle is an important parameter in SNMM measurements as probes with bigger conical angles contribute more to the stray capacitance. Utilizing the previous data about the final shape of the tip, we can calculate the expected value of the etching angle, which should be obtained by considering the length of the etched part of the tip ($L_{e}$) and the initial value of the pin diameter (*W*). We obtain approx. $atan\left(0.5\times W/L_{e}\right)=10$°, with a full aperture of approx. 20°, which is in quite good agreement with the obtained experimental result.

Figure 9 shows the SEM images of the fabricated tips. Figure 9a shows the tip fabricated with etching parameters of 2 mm/sec velocity with a 25% concentrated solution and Figure 9b shows the top view of the tip. Figure 9c shows the fabricated tip with a higher velocity. Due to the high moving speed, the control of the etching process was a little harder and the obtained tip has an apex diameter of more than 10 μm at 650 cycles, sharper tips can be obtained by increasing the number of cycles. Figure 9d shows the tip fabricated with a low concentrated solution, and, due to the slow process, the etching was completely controllable and the final tip shape was good with a tip apex size of less than 1 μm. A small perturbation in the process comes from the vibration induced by the stepper motor, and we expect that a shape with less surface roughness can be obtained by improving the mechanical control. By controlling the vibration which arises from the stepper motor, better control over the shape of the tip apex with less surface roughness can be achieved. The morphological analysis evinces that the tips with a micrometer and sub-micrometer apex sizes can be effectively obtained through this etching mechanism. This shows a remarkable agreement with the discussion of the etching process presented in the previous sections. Specifically, the final shape of the etched tip is consistent with what was previously described. Additionally, the chemical process used for etching was applied to the last 0.7 mm of the pin’s total length of 1.6 mm. Moreover, the width of the pin was reduced from 254 μm to ∼1 μm (approx.) after the etching process.

Using the previous equations to compare the experimental results with theoretical expectations leads to underestimating the expected etching rate. By studying the result shown in Figure 9d, where the measured end of the tip is lower than 1 μm, i.e., almost fully etched, the theoretically expected final width of the tip using the same process parameters should be obtained quickly. The etching rate estimated by inverting Equation (Equation 5) should be $r_{e}=0.17$μm/sec (approx.). Looking at the predicted etching speed, we should have a very slow etching process. Considering that we needed almost an hour (approx. 50 min) to complete the process, we have to account for a longer time of etching. Such a delay is justified by the necessary onset of the process, which does not start immediately. Still, it needs an activation time related to a few parameters: (i) concentration of the etching solution and (ii) temperature. Therefore, the etching rate as a function of time can be regarded as a sigmoid curve: it will exhibit a slow start, an increase, and a plateau, at least for reasonable times, assuming that the chemical solution is not changed over time owing to the small size of the tip respect to the volume of the etching liquid. To date, the value previously inferred can be considered the plateau of the sigmoid curve, similar to those presented in detail in Figure 4 of [35]. We have a lower rate because, in our case, no stirring of the solution has been included, to improve the quality of the etched surface. An evaluation of the time necessary for the complete etching of the pin can be done considering the results obtained in terms of the chemical solution efficiency. We already discussed the evidence for an effective etching when at least 300 dipping cycles are imposed at the dipping velocity of 2 mm/s. Since the pin is dipped for 0.7 mm, the dipping time for a single cycle is 2×0.7/2s=0.7 s. This can be considered equivalent to saying that the effective time to begin the process with the concentration 20% and at the room temperature is $t_{begin}=0.7$ s×300cycles=210s. The function for describing the etching up to the complete dissolution of the Cu apex of the pin can be written, accounting for the initial value of the diameter and for the normalization to 1 when the process is beginning, by using the inverse of a hyperbolic cosine, such as:(11)$$W\left(t\right)=\frac{2W}{exp\left(-\frac{t}{\tau }\right)+exp\left(\frac{t}{\tau }\right)}$$
where W(t) is the evolution in time of the pin width during the etching process, and the fitting parameter $\tau =2\times t_{begin}=420\;$s is the time constant of the curve helpful to account for the complete etching of the pin to get a sub-micrometric apex, as it is shown in Figure 10.

An additional contribution to be considered is the meniscus effect. It causes continuous contact between the tip and the liquid when the tip is very close to the solution’s surface. This problem has been regarded mainly in the scientific literature for studying the shape of the interface with various geometries, including droplets and cylindrical fibers dipped in a liquid solution [36]. The last case is coherent with our problem because the original shape of the tip is a cylinder dipped in the etchant solution. In the cited paper, the meniscus height is calculated by accounting for different ratios between the radius of cylinder *R* and that of the recipient *L* hosting the solution [37]. The significant case for our purposes is the case of a cylinder much smaller than the recipient, i.e., with L≫R. On the other hand, the radius of the cylinder is continuously changed owing to the etching process and the up-and-down movement of the tip, with the final result having a conical shape. To date, the findings in the literature should be updated to account for the reshaping of the tip during the etching process, even if the approximation L≫R is still valid, as *R* is decreasing over time. Looking now at the results for metals published in [38], the contact angle between the liquid surface and the cylinder ranges between $50^{\circ }$ and $80^{\circ }$, depending on the atomic number. Since we are using a Cu-Be alloy, we have *Z*(Cu) =29 and *Z*(Be) =4. The percentage of Be inside the alloy is typically limited to 3%, so we can assume that the equivalent value of *Z* for the alloy is very close to that of the Cu, and the experimentally determined contact angle will be $72^{\circ }$. Being R=W/2, where *W* is the initial diameter of the cylinder and *R* the radius, $\theta_{1}=72^{\circ }$ the contact angle, E=0.57721 the Euler–Mascheroni constant, and $\kappa^{-1}\approx 2.7$ mm the capillary length at $25{\;}^{\circ }$C, we can obtain a value for the meniscus height Δh≈125μm by using Equation (23) in [37]. This value fully justifies the final conical shape at the end of the etched pin.

Actually, a chemical process utilizing a precision mechanical setup for dipping the pin to be etched in the ferric chloride solution provides excellent control of the shape and the size of the tip apex. In fact, the final result of the etching is a tip end at the micrometric scale, which is a very good result for a chemical etching process. Looking at literature results, where mechanical or electrochemical techniques have been used to get sharp tips, we should compare our results with similar solutions to get small-size tips. The proposed technology lies between the aperture-ended probes, limited to an order of magnitude of 30 μm for waveguide solutions, as considered in past papers [39] and those to be used in AFM or STM setups for nano-characterization. The probes discussed in the classical literature already gave the opportunity for a μm size apex of the tip, since seminal papers such as [40]. In our contribution, we stress the possibility to control not only the size of the tip end but also the final shape by choosing the mechanical parameters governing the dipping process. In addition, because of the entire setup design, a quite large area can be scanned with such a high-resolution tip, because the scanning is performed by moving the sample with respect to the tip by means of precision step motors without contact between the tip and the sample.

### 3.2. Near-Field Analysis

To analyze the electric field concentration at the probe tip, a 2-D axis-symmetric finite element method (FEM) based electromagnetic simulations have been performed using the commercial solver COMSOL Multiphysics. The near-field characteristics of the probe tip have been analyzed mainly for two different categories: (1) flat edge tip with sharp edges and (2) blunt edge tip with smooth edges. By observing the tip morphology from the SEM micrograph, Figure 9 reveals that the fabricated tips are mostly having a blunt edge conical shape tip.

The electric field map of the two category probes along with the unetched standard cylindrical pin with a radius of 125 μm at 6.5 GHz is displayed in Figure 11. Simulations were carried out utilizing the Comsol RF module, incorporating scattering boundary conditions imposed on the domain walls of the simulation boundaries. Given the geometry of the model, which includes details ranging from mm size to sub-micron levels, a suitable meshing was executed with custom element size parameters. The electric field is concentrated at the sharp edges of the flat tip but on the other hand, the blunt edge tip shows uniform and symmetric fields around the tip. The sharp edge of the unetched pin has a concentrated field but with an FWHM (Full Width Half Maximum) of more than 120 μm. The fields are still effective up to 10 μm in distance from the tip and get higher when reaching 1 μm. At 10 μm distance, both tips are yielding a FWHM of around ∼50 μm and at 1 μm distance this reduces to 4 μm. This demonstrates that the tip-sample distance is very important in getting a good sensitivity in receiving the reflected signals within this range. The blunt edge tip also has reasonable focusing of the fields compared to the flat edge tip and this will result in increasing the measurement spatial resolution. This also validates that these tips can be able to provide a good resolution in the order of a few microns when imaging with these tips.

### 3.3. SNMM Imaging Analysis

SNMM imaging has been performed using the same setup described in [41], with a motorized test fixture to move the sample with respect to a fixed tip, vertically positioned at a short distance from the surface of the sample. Fixing the tip and moving the sample corresponds to an improvement in the measurement stability, owing to the small power collected as a reflected signal. In fact, there are significant signal losses along the travel on the air from the tip to the sample and back to the tip. The setup is designed to have a non-contact measurement, with the tip scanning over the sample at a distance of a few μm. Since the tip is rigid, this solution allows a stable and sensitive measurement without damaging the sample surface or producing any wearing to the tip during the scanning procedure. In fact, the tip is sensitive enough to receive back a non-negligible amount of reflected power even without touching the surface, but it is also small enough to maintain a high resolution of the measurement even over an extended area of characterization. Amplitude and phase measurements ($S_{11}$) of the reflected signal have been performed using a vector network analyzer (VNA) to compare the results of both data. In [42], the criterion to choose a measurement at resonance or off-resonance (but close to it) has been discussed, with the aim to enhance the picture’s contrast for imaging purposes. The setup includes phase shifters and resonance elements. This solution does not allow an instant broadband measurement, but it is ideal to have high-quality and well-contrasted images at frequencies determined by the multiple resonances of a λ/2-resonator. The result is shown in the images of Figure 12, for metal planar shapes manufactured by photolithography of gold deposited on an alumina substrate. A virgin pin Figure 12a has been used as a reference, comparing the measurement without etching of the pin with the images obtained using the amplitude Figure 12b and the phase Figure 12c of the reflected signal obtained with an etched micrometric pin. It is evident the quality improvement when passing from Figure 12a to Figure 12b,c. The higher $\left|\Delta S_{11}\right|$ of Figure 12a compared to Figure 12b is attributed to its bigger tip area available to receive the reflected signal from the sample, which is a trade-off between the sensitivity and the resolution [7,30]. As in this case, a non-contact mode of operation is considered, and any issues related to the wearing of the tip can be neglected.

The main differences between the two shapes of the probes used for obtaining the imaging of the patterned Au on the alumina substrate are the size of the end of the probe and the potential contribution of a stray capacitance coming from the vertical wall of the tip. Despite the absence of cantilevers to host the tip, which causes significant stray contributions in AFM setups, the vertical wall of the virgin cylindrical pin and the reshaped conical tip obtained after the etching have both possible lateral contributions of the EM field. On the other hand, as well-established in the literature [5], when the end of the tip is very small with respect to the total length this causes a charge accumulation close to the end, with smaller or negligible lateral EM contributions. For the above reason, the resolution of the measurement technique is mainly determined by the size of the end of the tip, as evidenced in Figure 12. The measurement resolution of our scanning setup is determined by both the tip apex size and the accuracy of the step motors controlling the movement of the sample below the probe. The precision of the latter can be improved through the utilization of more advanced piezo scanners. Considering that the currently available step-motors in our setup allow a minimal movement close to 2.5 μm, this value can be considered the best result in resolving the details of the scanned samples.

## 4. Conclusions

In this paper, we demonstrated the improvement in the imaging resolution of a near-field homemade microwave microscope utilizing ultra-sharp probes obtained by dynamic chemical etching of the microwave broadband flange connectors. The spatial resolution of the near-field imaging system is in the order of the tip apex size, and it is necessary to have a sharp tip as a probe to get improved images at the micrometric scale. The method yielded tapered tips with tip apex sizes of ∼1 μm with a small cone angle between $10^{\circ }$ to $15^{\circ }$, which is necessary for the SNMM in order to reduce any stray/parasitic capacitance. The optimization of the chemical etching technique has been performed by varying a number of parameters: changing mainly the dipping velocity and the concentration of the etchant solution. The near-field characteristics of these fabricated probes showed the focusing ability of these probes that are in the order of a few microns and which is good enough for a microscopy system to characterize samples that require micron and sub-micron resolution and scanning areas of several mm${}^{2}$. In addition, these probes show good radiation characteristics, thus enhancing the resolution of the measurements. The probes are designed to be operated in the non-contact mode, but this is expected because of the completely different design approach of the setup with respect to commercial AFM-based solutions. Actually, the main goal is to build up a different system, where a robust, microwave-efficient tip is used for studying sub-surface, buried details in a non-destructive way. Since the interface to the microwave components is mostly provided by the SMA connectors, these probes also offer ease of integration with microwave feeding components of the SNMM setup to make a proper connection with VNA. The measurement sensitivity can be further improved by incorporating resonators with the developed probe. In the end, these probes are showing promising results and are eligible to be used as effective probes for a microwave microscopy system.

## Figures and Tables

**Figure 1 sensors-23-03360-f001:**
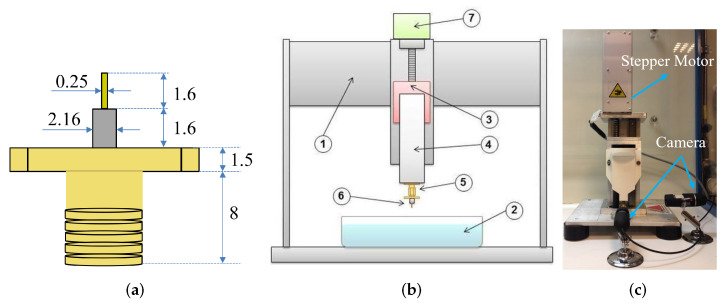
Experimental setup. (**a**) Geometry of the SMA flange connector, all the dimensions are in mm; (**b**) schematic of the experimental arrangement (1. Mechanical frame, 2. Solution container, 3. Movement assembly, 4. Moving stage, 5. SMA Male Plug, 6. SMA female connector, 7. Stepper motor); (**c**) photograph of the experimental arrangement.

**Figure 2 sensors-23-03360-f002:**
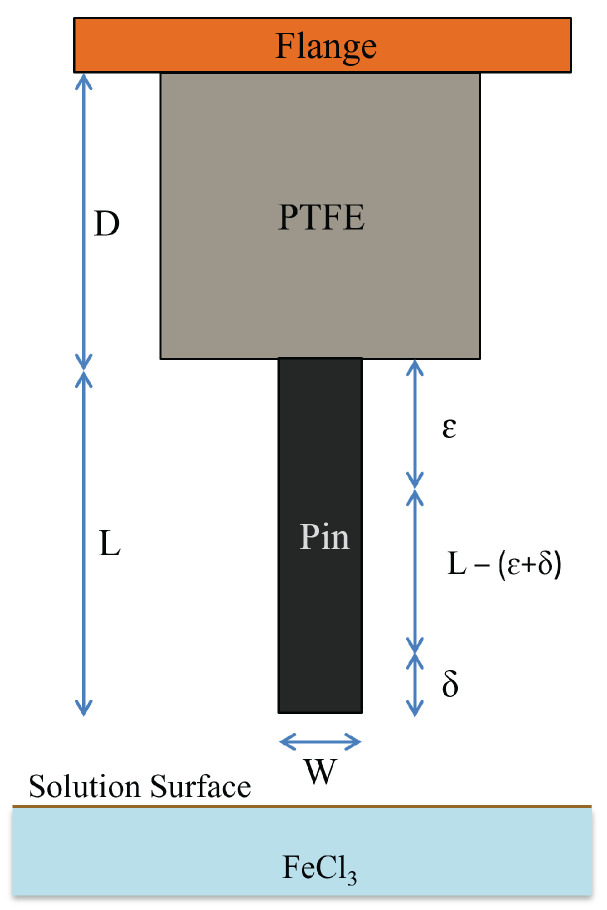
Schematic of the flange connector to be etched (not in scale).

**Figure 3 sensors-23-03360-f003:**
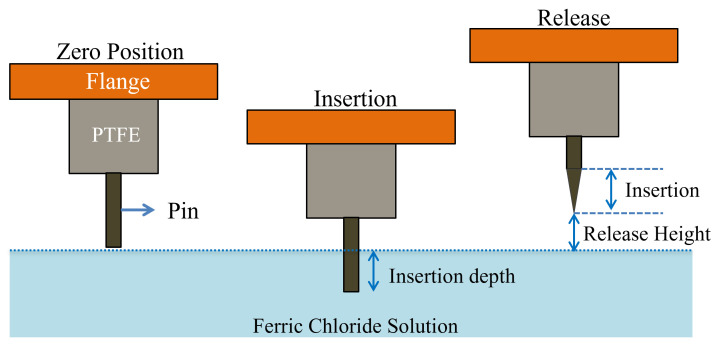
Schematic diagram of the dynamic etching process.

**Figure 4 sensors-23-03360-f004:**
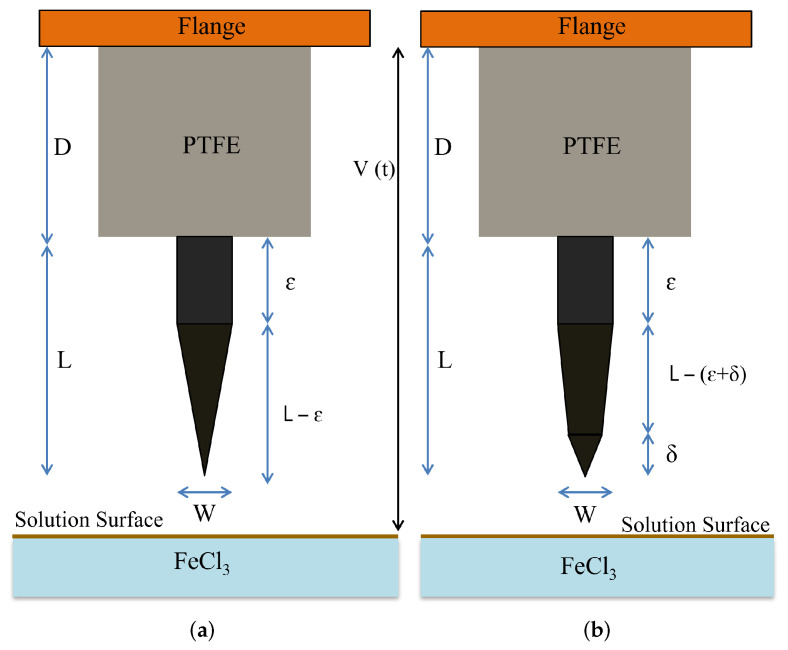
The final shape of the etched pin is qualitatively shown accounting for the two possible results: (**a**) meniscus effects are not present, (**b**) meniscus effects considered.

**Figure 5 sensors-23-03360-f005:**
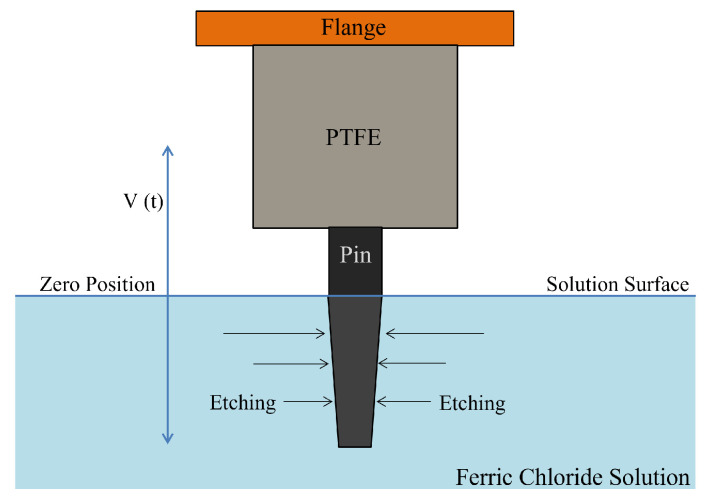
Reshaping of the pin during the etching process.

**Figure 6 sensors-23-03360-f006:**
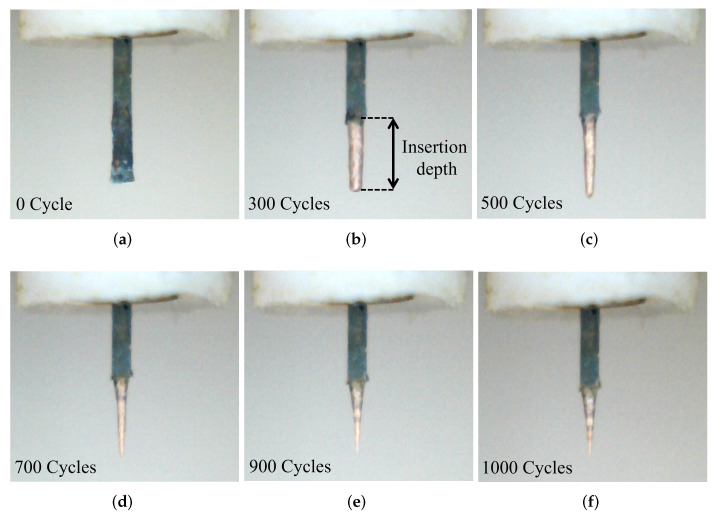
Reshaping of the pin during the etching process: (**a**) starting shape, (**b**) after 300 cycles, (**c**) after 500 cycles, (**d**) after 700 cycles, (**e**) after 900 cycles, (**f**) final shape after 1000 cycles.

**Figure 7 sensors-23-03360-f007:**
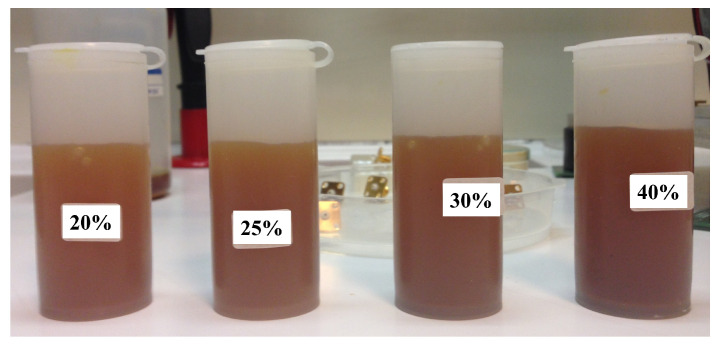
Photograph of the prepared solutions with different concentrations.

**Figure 8 sensors-23-03360-f008:**
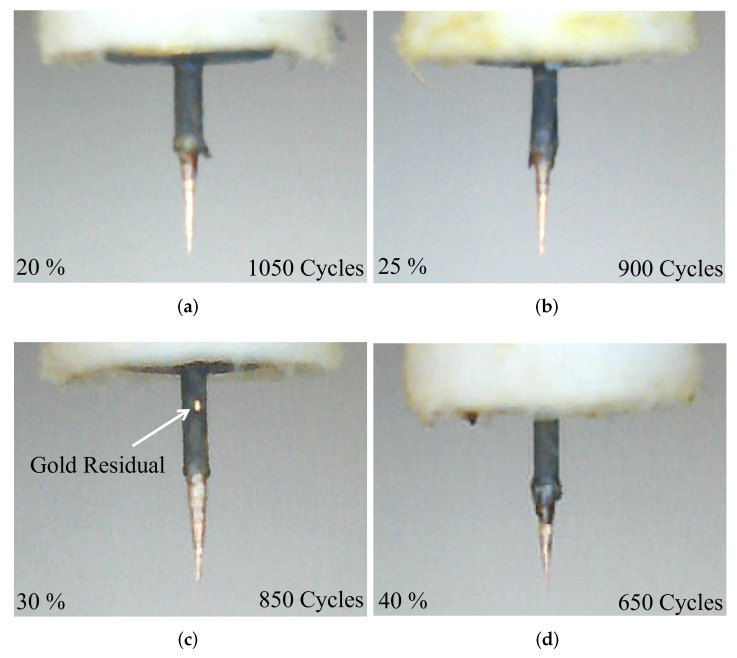
Etching process results with different concentration solutions: (**a**) 20% solution, (**b**) 25 % solution, (**c**) 30% solution, (**d**) 40% solution.

**Figure 9 sensors-23-03360-f009:**
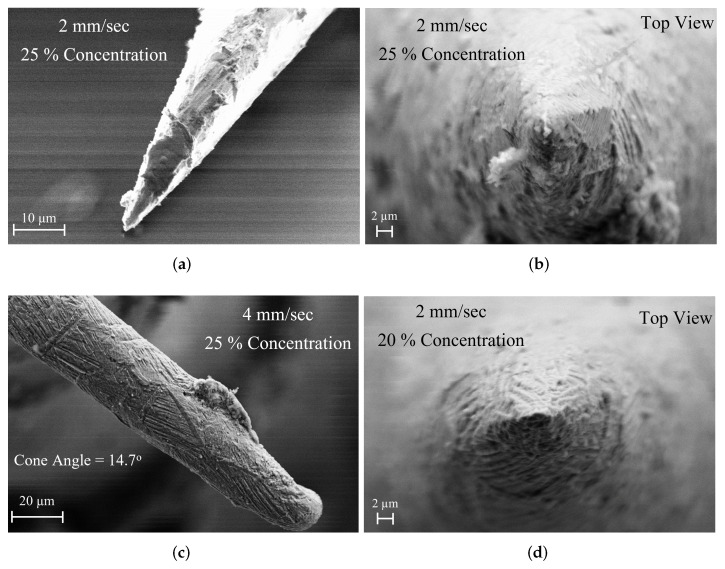
SEM micrograph images of the fabricated tips. (**a**) 2 mm/s velocity with 25% concentrated solution, (**b**) Top View of the tip prepared with a velocity of 2 mm/s with 25% concentrated solution, (**c**) 4 mm/s with 25% concentration, (**d**) 2 mm/s with 20% concentration.

**Figure 10 sensors-23-03360-f010:**
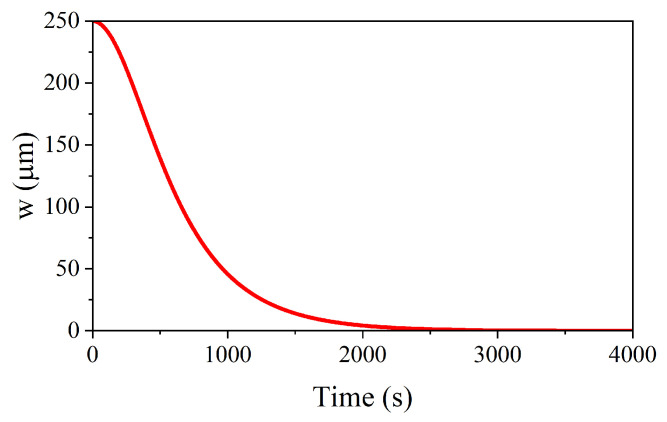
Dynamic etching of the pin starting from the initial value of the diameter (250 μm) down to the micrometric size. The complete etching is obtained after 3000 s = 50 min (aprox.).

**Figure 11 sensors-23-03360-f011:**
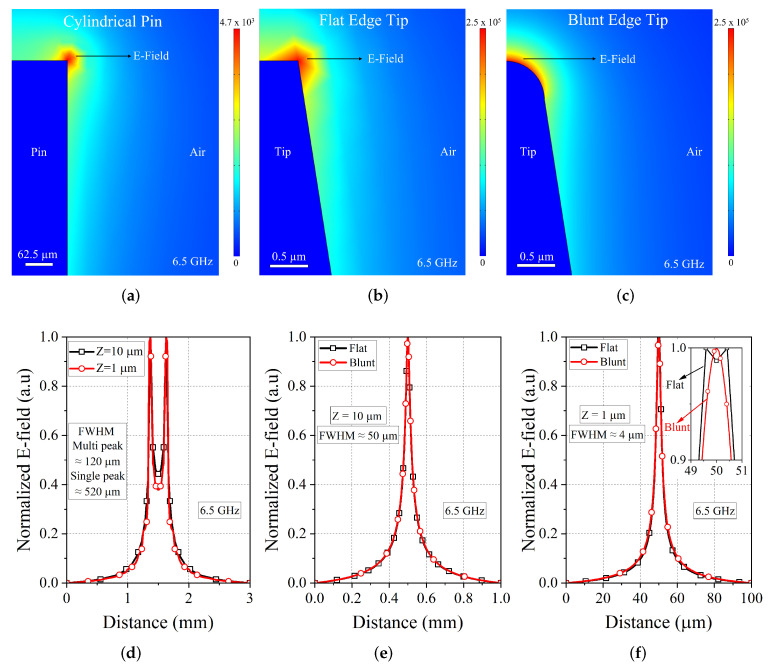
Simulated electric field map of the probes. (**a**) cylindrical pin; (**b**) flat edge tip; (**c**) blunt edge tip. (**d**) Normalized E-field profile of the cylindrical pin taken at two different distances from the pin. The field profiles of the flat and blunt edge conical tapered tips were taken at a vertical distance of (**e**) 10 μm and (**f**) 1 μm.

**Figure 12 sensors-23-03360-f012:**
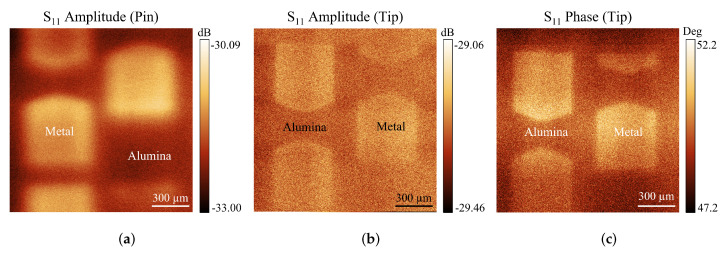
Imaging with the SNMM of gold-patterned shapes obtained on an alumina substrate by photolithography. In (**a**), the image has been derived by an ordinary pin of a microwave flange connector; in (**b**,**c**), amplitude and phase have been measured using the signal reflected by the sample back to an etched micrometric tip.

**Table 1 sensors-23-03360-t001:** Parameters of the velocity optimization.

Velocity (mm/sec)	Total no. of Cycles	Time for 100 Cycles (min)	Insertion Depth (mm)
1	1000	10:59	0.5
2	1000	4:59	0.8
4	1000	3:42	0.7

**Table 2 sensors-23-03360-t002:** Optimization parameters of the solution concentration.

Concentration (%)	Total no. of Cycles	Velocity (mm/sec)	Time for 100 Cycles (min)	Insertion Depth (mm)
40	650	2	4:59	0.7
30	850	2	4:59	0.7
20	900	2	4:59	0.7
10	1050	2	4:59	0.7

## Data Availability

Not applicable.

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
