# Peer review of "Fabrication of Ultra-Sharp Tips by Dynamic Chemical Etching Process for Scanning Near-Field Microwave Microscopy"

_sensors, 2023, doi:10.3390/s23063360_

Round 1

Reviewer 1 Report

An effective dynamic chemical etching technique to fabricate ultra-sharp tips for Scanning Near-field Microwave Microscopy is designed in this paper. This work is highly innovative and has certain practical value. This work is innovative and has certain practical value. However, there are still some minor problems to be revised:

1. Section 3.2, what are the boundary conditions of this simulation? Is it in a confined space or an open space? What is the input condition?

2. Section 3.2, how is the mesh setting in Figure 11? Could you provide a mesh structure diagram? Since this is a small model, the unreasonable mesh setting may also cause the wrong focus of the electric field.

3. Section 4, how to explain “These probes also offers ease of integration with microwave feeding components of SNMM setup to make a proper connection with VNA”?

Author Response

see attached document

Reviewer 2 Report

This paper reports the facile fabrication of SNMM tip by an etching process. The authors developed a mechanical device to control the dynamic etching process where a pin is inserted into a ferric chloride solution. The sharp tip was achieved by optimizing the insertion program and the concentration of the ferric chloride solution. The processed tip exhibited higher resolution in SNMM imaging of gold-patterned shapes. Here are the comments to the authors.

(1) What is the definition of “Ultra-sharp” tips?

(2) Please compare the sharpness of the tips quantitatively with the previous reports.

(3) How the pin was fixed for SNMM operation. It seems like the pin is made of flexible material. Does it affect the spatial resolution of SNMM?

(4) Please evaluate the sharpness of the pin quantitatively. Only photographs and SEM images are indicated, and these are qualitative data.

(5) In the SNMM image, only the pin and tip were compared. The factors such as the shape of the tips should also be compared.

(6) Please indicate the resolution range by the current tip. 
